# Pseudo-spin–valley coupled edge states in a photonic topological insulator

Yuhao Kang [1,2], Xiang Ni [2,3], Xiaojun Cheng[1,2], Alexander B. Khanikaev [2,3] & Azriel Z. Genack[1,2]

Pseudo-spin and valley degrees of freedom engineered in photonic analogues of topological insulators provide potential approaches to optical encoding and robust signal transport. Here we observe a ballistic edge state whose spin–valley indices are locked to the direction of propagation along the interface between a valley photonic crystal and a metacrystal emulating the quantum spin–Hall effect. We demonstrate the inhibition of inter-valley scattering at a Y-junction formed at the interfaces between photonic topological insulators carrying different spin–valley Chern numbers. These results open up the possibility of using the valley degree of freedom to control the flow of optical signals in 2D structures.

[1] Department of Physics, Queens College of the City University of New York, Flushing, NY 11367, USA. [2] The Graduate Center of the City University of New York, New York, NY 10016, USA. [3] Department of Electrical Engineering, Grove School of Engineering, The City College of the City University of New York, New York, NY 10031, USA. Correspondence and requests for materials should be addressed to A.B.K. (email: akhanikaev@ccny.cuny.edu) or to A.Z.G. (email: genack@qc.edu)

Topological concepts have recently entered the realm of photonics. Topological states of light engineered in a variety of systems, from magnetic photonic crystals to silicon ring resonators and waveguide arrays, and across the electromagnetic spectrum, from the microwave to the optical domains, have been shown to exhibit fascinating phenomena such as robust guiding and quantum entanglement of photons[1–9]. These demonstrations and the great potential for photonics applications are spurring research in the optical domain. However, one significant problem in scaling down topological designs is the complexity of their geometries, which make them hardly feasible with present-day nano-fabrication techniques. There has therefore been growing interest in simpler designs in which topological states could be emulated in photonics by using lattice symmetries and associated synthetic degree of freedom (DOF) that might be achieved in structures that are less challenging to fabricate. One such approach is based on the use of the valley DOF. Valley is an additional discrete synthetic degree of freedom in crystals with triangular and hexagonal point symmetries. Valley refers to one of the two high-symmetry points of the Brillouin zone, the $K$ and $K'$ points and their immediate neighborhoods, and can be viewed as a pseudo-spin DOF. Conservation of the valley DOF under a broad class of perturbations[10] makes it suitable for emulating photonic topological states and facilitates the design of valley Hall photonic topological insulators (VHTIs)[10–16].

The valley DOF has recently gained prominence in condensed matter physics in the context of valleytronics (i.e., valley-locked electronic propagation) in a wide variety of materials[17–21]. In VHTIs, one considers a restricted topological phase of photons that is defined at only one of the two valleys and is characterized by a valley projected half-integer Chern number associated with the valley: $C^{K/K'} = \pm 1/2$. Such a restricted topology is often sufficient to lead to edge states at the domain walls between parity conjugate VHTIs with opposite valley Chern number[11]. To distinguish such edge states from states of conventional electronic and photonic topological insulators (TIs), such edge states are also referred to as valley–Hall kink states[22–26].

The control of currents by manipulating the valley DOF has been successfully realized in novel electronic devices such as the valley splitter and the valley valve[27–29]. These developments have further stimulated the exploration of the valley DOF of photons. The interest in generating fully valley-polarized electromagnetic waves stems from potential applications in valley-based information encoding and processing[30,31]. However, in order to manipulate the valley DOF, it is necessary to properly break the valley degeneracy. For instance, a photonic valley splitter was achieved by utilizing the valley-dependent trigonal warping distortion in graphene bands[28,32]. Previous publications have suggested that a valley photonic crystal (VPC) could be created by lifting inversion symmetry[10,33]. These results show that valleytronics may provide a practical approach for realizing a full control of photonic states in valley–Hall systems.

The development of photonic TIs opens up the possibility of extending valleytronics to optics where the valley DOF is combined with the pseudo-spin DOF responsible for the topological order. Although interest in the valley DOF in the context of topological photonics is increasing rapidly, the properties of such hybrid topological states combining valley-polarized waves with other synthetic degrees of freedoms in photonic TIs have not been realized experimentally.

Here we demonstrate that the combination of valley and pseudo-spin degrees of freedom in one system enables unidirectional states along the interface between aVPC and a photonic TI[34,35] with entangled valley and pseudo-spin degrees of freedom. Based on this property, we construct a spin-valley polarized splitter for the edge states which can be used to route signals in optical networks and interconnects.

## Results

**Metacrystal design.** To create valley-polarized edge states confined to an interface, we juxtapose a VPC and a topological

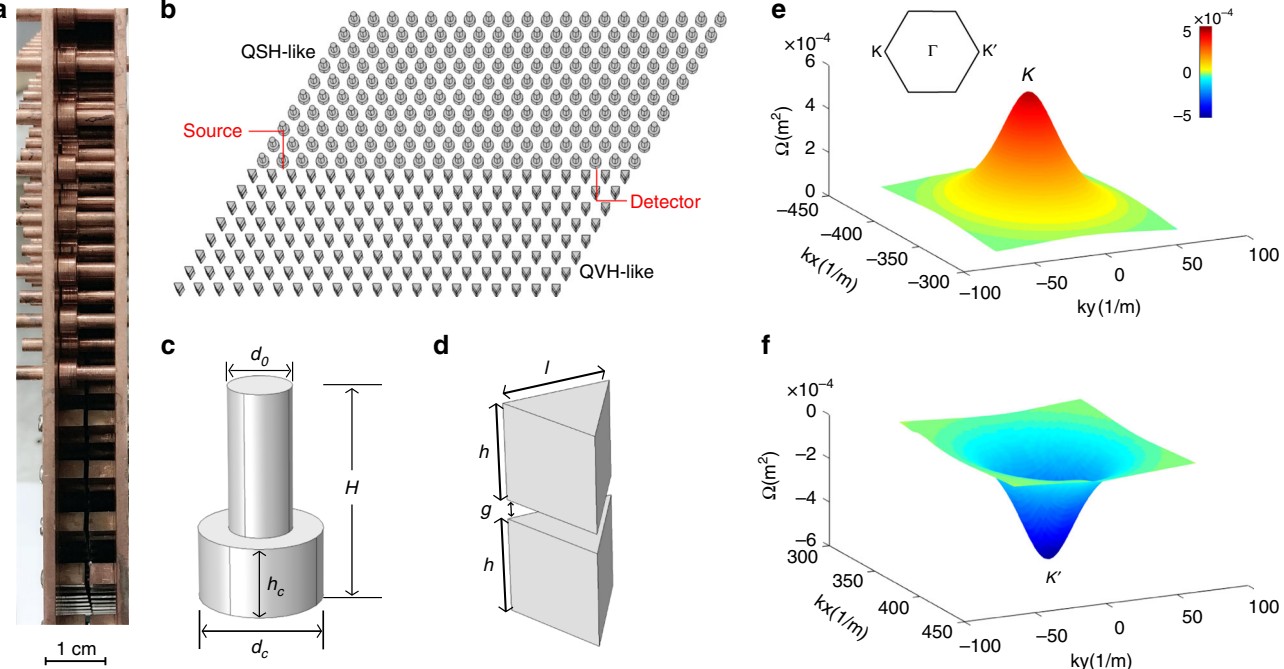

**Fig. 1** Schematic of the sample configuration. **a** Side view of the structure. **b** Arrangement of two crystalline regions with the two copper plates on top and bottom removed. Lattice constant $a = 1.0890$ cm. **c**, **d** Unit cells of two crystals emulating the quantum spin–Hall (QSH) and quantum valley-Hall (QVH) effects. Dimensions of unit cells: $H = 1.0890$ cm, $h_c = 0.3580$ cm, $d_c = 0.6215$ cm, $d_0 = 0.3175$ cm, $h = 0.5040$ cm, $g = 0.0810$ cm, $l = 0.5020$ cm. **e**, **f** Berry curvatures of the triangular lattice (valley Hall crystal) for the TE mode in the valence bands near the K and K' points

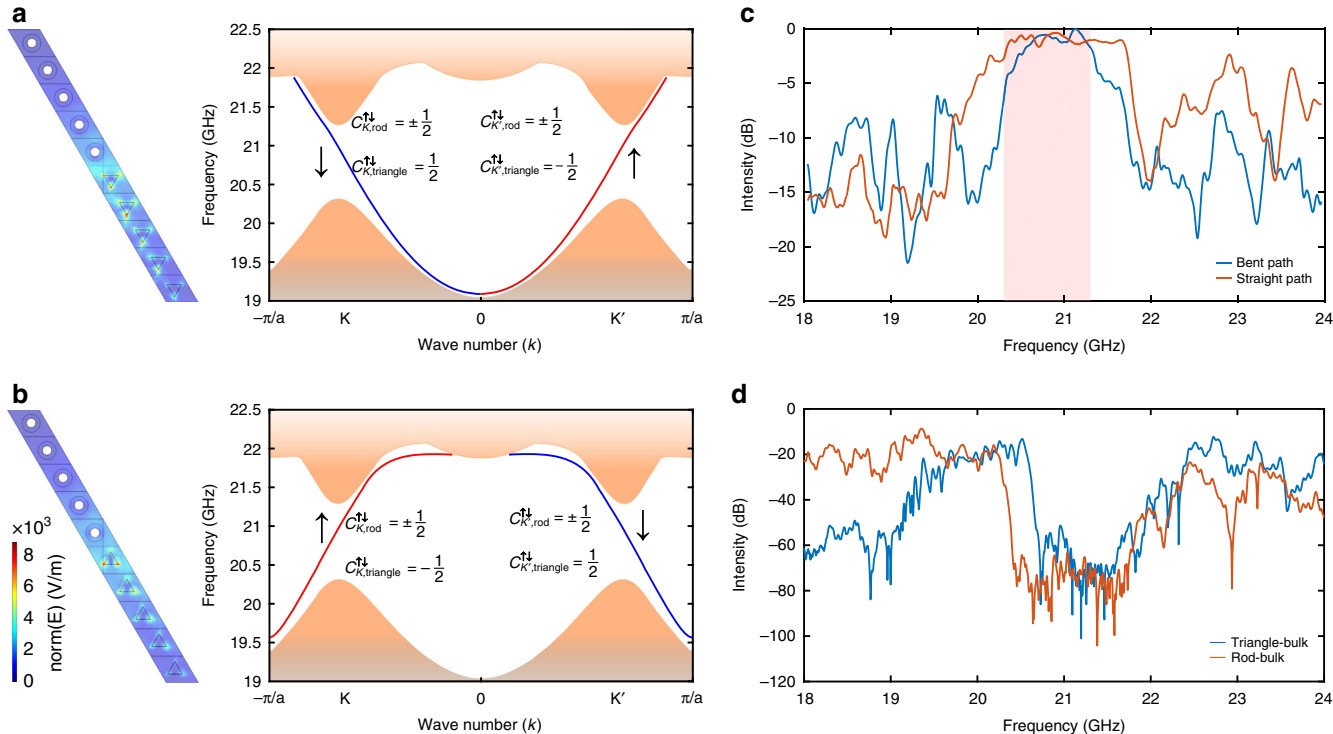

**Fig. 2** Edge states between valley–Hall and quantum spin–Hall domains. **a**, **b** Edge band diagram of a 40 × 1 supercell, with an interface between the rod-collar-lattice with collars in contact with the lower plate and triangle-lattice. There are two gapless edge states which correspond to pseudospin-up (↑, red line) and down (↓, blue line) states at the K′ and K points, respectively. The bandgap spans the frequency range from 20.3 to 21.3 GHz. **c** Transmission spectra along the straight and bent paths. The shaded region indicates the bandgap. **d** Transmission spectra through the bulk of the rod and triangle lattices

photonic crystal (upper region in Fig. 1a, b) possessing a complete topological band gap. The elements of the topological crystal are copper rods with concentric collars, as shown in Fig. 1c. The rods can be pushed to be in contact with one or the other of the bounding copper plates or to lie between the plates. When the collar deviates from the midpoint between the plates, $\sigma_h$ symmetry is broken and a bandgap opens at Dirac points. Reducing $\sigma_h$ symmetry leads to coupling between transverse-electric-like (TE) and transverse-magnetic-like (TM) modes, in which the $E_z$ and $H_z$ components vanish. The eigenstates of the structure are then a mixture of TE and TM modes whose in-phase and out-of-phase combinations define pseudo-spin up and down states[36,37]. This rod-collar-lattice emulates the quantum spin–Hall (QSH) effect[38,39], with the bulk bands (with collars at the bottom, Fig. 1c) acquiring spin-Chern numbers $C_{\uparrow\downarrow} = \pm 0.5$ in both the K and K′ sectors[36] (The effective Hamiltonian is shown in Supplementary Note 1).

The structure[10] emulating the quantum valley Hall (QVH) effect is shown in the lower region of Fig. 1b. Copper triangular prism pairs with a gap between them (Fig. 1d) are arranged in a triangular lattice. A bandgap appears for this orientation of the triangles[40], as shown in the bulk photonic band structure (Fig. S1 in ref. [34]). The dimensions of the triangular prisms are selected so that there is an appreciable overlap of the bandgap of the rod-collar-lattice and the triangle-lattice. As a result, electromagnetic waves propagating in the metawaveguide within this frequency range are confined to the interface and decay exponentially into the surrounding domains. A novel feature of this interface is that the edge states are spin–valley-polarized, providing an additional tool for guiding transport as compared to a trivial 1D channel.

Unlike the rod-collar-lattice with bianisotropic response, the TE and TM modes in the triangular structure are decoupled

owing to the $\sigma_h$ and $C_3$ symmetry of the triangular prism[11]. The structure is designed so that TE and TM modes are degenerate near the K/K′ valleys and the pseudo-spin states can still be defined. The triangular unit cell differs from a trivial photonic crystal in that it breaks inversion symmetry. This gives rise to different local Chern numbers in the two valley sectors. The Berry curvature[41,42] for the TE mode around the two valleys is shown in Fig. 1e, f. The local valley Chern numbers in the K/K′ sectors are ±0.5, while the global Chern number vanishes due to TR symmetry. Similar results are obtained for the TM mode as a result of the degeneracy of the TE and TM modes. These numerical results agree with the theoretical value obtained using the effective Hamiltonian method[10]. We note that rotating the triangular prisms by 180° reverses the valley Chern number for the TE/TM modes from $C_{K/K'} = \pm 0.5$ to $C_{K/K'} = \mp 0.5$[11]. The bulk-interface correspondence principle ensures the presence of edge states when there is a difference in the topological invariant across the interface[43–45]. As a result, the Chern number difference is $(-0.5) - 0.5 = -1$ for the super-cell shown in Fig. 2a in the spin-down sector of the K valley. This corresponds to a backwards propagating spin-down state at the K valley. Similarly, a forward spin-up state exists at the K′ valley. These counter-propagating edge states are protected by TR symmetry and are immune to backscattering[46,47] in the absence of a magnetic field and magnetic materials.

**Edge states and effect of disorder**. As a first step, we demonstrate the existence of edge states by measuring the transmission spectrum. Measurements are performed for a zigzag cut of the crystal, as shown in Fig. 1b, because reflection of the edge state at the interface between the metawaveguides and air is inhibited in this case[10]. The source and detector dipoles are inserted vertically

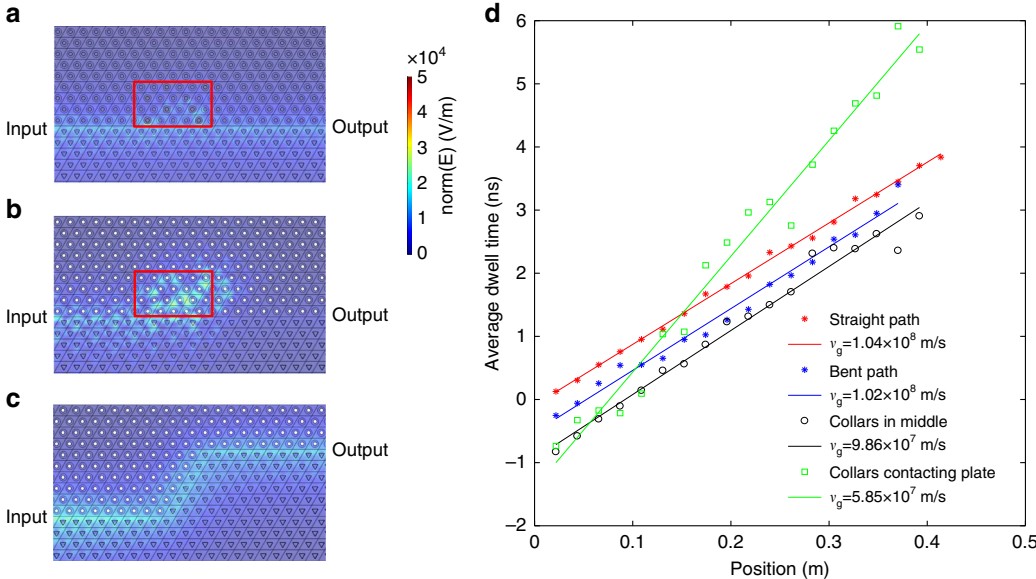

**Fig. 3** Propagation in the presence of disorder. Ten randomly selected rods are perturbed inward or outward relative to the plane of the figure in a small region (red box) near the boundary in each random configuration. **a–c** Simulation results. **a** The collars on the pushed rods are not in contact with either copper plate. Waves passing through the disordered region continue to propagate in the forward direction. **b** The rods are moved so that the collars touch the opposite plate. The wave is strongly backscattered so that a speckle pattern develops due to the interference of scattered waves. **c** Transmission profile in the bent path. **d** Experimental results. Comparison of the dwell times along the domain wall in four cases. The group velocity $v_g$ is the inverse of the slope when propagation is ballistic. Negative values of $\tau(\omega,x)$ indicate that the pulse is reshaped in the limit of zero bandwidth

through small holes in the copper plate. A comparison of transmission spectra for a straight line and a path with a 60-deg turn (Fig. 3c) is shown in Fig. 2c. Although the intensity near the left edge of the bandgap (~20.3 GHz) decreases by 5 dB in the case of the bent path relative to the intensity for the straight path, the signal at the middle of the bandgap is the same for both paths. The high-transmission plateau inside the bandgap (shaded part in Fig. 2c) reflects the confinement of the edge state. This contrasts with a drop in transmission of ~15 dB in the pass band. In addition, the strong suppression of transmission in the bulk of the TI lattice and the lattice of triangles, which is seen in Fig. 2d, confirms that the observed transmission is due to the excitation of the edge state. This proves the robustness of the edge state encountering a bent path. Strong suppression of backscattering at a corner is a signature of topologically protected edge states and makes it possible to realize unidirectional electromagnetic transport along a complex path.

Structural defects are inevitable in fabricated materials. Considering that the spin and valley indices of the states are locked to the transport direction, the edge state reflected at a turning point or at defects will experience both inter-valley mixing and spin flipping. Because of the large separation in momentum space between $K$ and $K'$, it is reasonable to suppose that the valley index is conserved to a high degree. Here, we focus on the spin–orbit coupling term, $H_{coupling} = \varepsilon m_B \hat{s}_z \hat{\tau}_z \hat{\sigma}_z$; pushing rods to the opposite copper plate switches $\varepsilon$ from 1 to −1. Two kinds of disorder are introduced in experiment: firstly, weak disorder when a collar is positioned between the plates so that $\varepsilon$ is between −1 and 1, and secondly, strong disorder when a collar is moved so that it is in contact with the opposite plate, so that $\varepsilon = -1$.

The strength of backscattering and the conservation of valley in the presence of disorder can be determined from measurements of the delay time. The band structure (Fig. 2a) gives a nearly constant value of the group velocity across the band gap, $v_g = \frac{\partial \omega}{\partial k}$ of $1.06 \times 10^8$ m/s. The group velocity can be extracted from the linear increase of the delay time with position along

the domain wall. The delay time is equal to the spectral derivative of the phase[48], $\tau(\omega,x) = \frac{d\varphi(\omega,x)}{d\omega}$, where $\varphi$ is the phase shift of the electric field between the source and detector, $\omega$ is the angular frequency, and $x$ is the distance from the source. The average delay time $\tau$ is obtained from an average over seven configurations.

The inverse of the slopes determined from the data in Fig. 3d for straight or bent paths are in agreement with the calculated group velocity. This demonstrates that the wave propagates ballistically along the bent paths. When some of the collars are positioned between the two plates, the measured velocity is found to be $9.86 \times 10^7$ m/s, which is close to the value of $1.04 \times 10^8$ m/s in the ordered sample. This shows that valley and spin are conserved on the scale of the metawaveguide used in the experiment. However, when disorder is introduced by having the rods touch the opposite plate, the mean dwell time increases (green line in Fig. 3d) compared to the dwell time for the edge state of the ordered crystal. This indicates that the wave follows longer trajectories because of scattering (Fig. 3b) and the spin–valley indices are not conserved. These measurements reveal the limits of robustness of the edge states.

**Valley-dependent waveguiding**. For photonic communications, the valley DOF can be used to encode a binary logic, 0 and 1. A valley splitter will then play an important role in filtering signals by their binary state. To achieve such functionality, we incorporate three different domains in a single platform: lattices with collars up or down (collars in contact with the upper or lower plate) and a triangle-lattice. The resultant Y-junction provides a basis for realizing the valley splitter[11,16]. The wave is injected into port 1, as shown in Fig. 4a. Two spin-down states at both valleys ($\Psi^{\downarrow}_{K/K'}$) are supported along the transport direction in the channel between collars-up and collars-down lattices[34], while each output channel of the Y-junction supports only a single spin-down state in a single valley ($\varphi^{\downarrow}_{K/K'}$). According to the photonic band structure (Fig. 2a, b), the edge

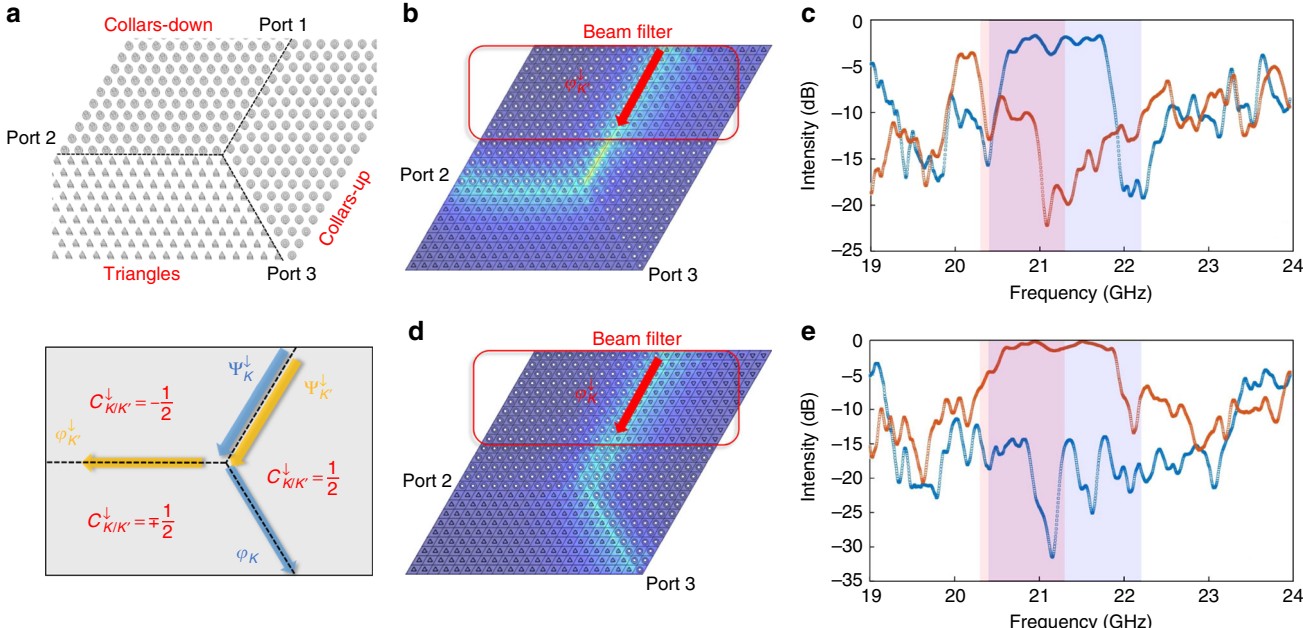

**Fig. 4** Topological Y-junction operating as valley filter with valley-splitting wave transport. **a** Schematic of the Y-junction formed by three domains. Energy is injected via port 1. Ports 2 and 3 only support K′ and K-polarized waves, respectively. The spin down state is indicated by the subscript ↓. The lower panel shows Chern numbers in three regions and the supported edge modes along the direction of the arrows. **b**, **d** The region in the red box functions as a beam filter that only passes signals carrying specific valley index. In **b**, the triangles inside the red box are upright, so that the $\varphi_{K'}^{\downarrow}$ state can propagate freely in the direction of the arrow. **d** When these triangles are inverted the beam filter only supports the $\varphi_K^{\downarrow}$ state. All transmission profiles are simulated at 21 GHz. **c**, **e** Comparison of experimental valley-dependent transmission from port 1 to ports 2 and 3. Blue and red lines indicate the intensities at port 2 and port 3, respectively. Light red and light blue shaded regions indicate the bandgap of the rod-triangle region and collars-up-down region

state $\varphi^{\downarrow}$ is exactly at the K/K′ valley at 21 GHz. For the $\Psi^{\downarrow}$ state, this frequency should be 21.5 GHz (Fig. 2 in ref. [34]). Since the pseudo-spin DOF does not determine the path here, the edge state will follow a path based on the valley polarization, provided that inter-valley scattering can be neglected at the Y-junction. To test this, we control the valley index of the source and compare the transmission spectra at the two outputs. In order to generate a single-valley signal at port 1, the wave first goes through a filter (a path segment between a rods-down lattice and a triangle-lattice) so that all incoming modes are evanescent except for the edge mode. The valley of the input edge state can be altered by rotating the triangles 60°. (from Fig. 4b–d). In the case of Fig. 4b, the input mode carries K′ valley. K′/K valley edge states are supported along the domain wall from the intersection towards ports 2/3 , respectively. If the valley is conserved at the intersection point, the wave should only follow the path to port 2. As expected, we observe a drop in transmission of ~10 dB around 21–21.5 GHz at port 3 (red line in Fig. 4c). As the frequency is shifted from the 21 to the 21.5 GHz range, the signal in port 3 remains small compared to the high transmission plateau of port 2 (blue line in Fig. 4c). This shows that the inhibition of inter-valley scattering is strongest for the edge states exactly at K/K′, i.e. the waves largely maintain their original valley indices. To further demonstrate this, we conduct a contrast experiment. When the valley of the input signal is changed to K (Fig. 4d), we see a similar phenomenon with port 2 and 3 exchanging roles (Fig. 4e). The dip of the signal in port 2 around 21.2 GHz is ~30 dB lower than the signal at port 3, which demonstrates the suppression of transport along the path to port 2. Simulations carried out with COMSOL Multiphysics are in agreement with measurement. In this experiment, the path selected is purely decided by its valley property. Distinct valley signals are guided

to different outputs with contrast of 20–30 dB. This indicates that the valley DOF of electromagnetic waves is robust at the Y-junction.

## Discussion
The experimental results presented here demonstrate the robustness of the edge states along the interface between QSH-like and QVH-like photonic crystals. Energy entering the system can move along a bent path without appreciable scattering or additional delay. The Y-junction integrated into such a system can then serve as a fault-tolerant valley splitter. While previous bulk valley splitters[28,33] could only steer valley-polarized waves in fixed directions (such as the ΓK or ΓK′ directions), the Y-junction demonstrated here is reconfigurable and the shape of the edge can be laid out in forms that are not restricted by the orientation of the unit cell of the 2D platform.

The approach taken here of exploiting pseudo-spin and valley DOF to guide microwave radiation can potentially be extended to the optical domain. In Supplementary Note 2, we present a design for a germanium nanostructure lattice that emulates the QSH effect. Designs for dielectric lattices supporting valley are also being pursued to create photonic structures in which the valley DOF can be used to control the flow of optical signals along domain walls separating QSH and valley photonic crystals.

## Methods
**Numerical simulations of Berry curvature and band structure.** The Berry curvature is calculated using the numerical eigenstates obtained with use of the COMSOL RF module. We apply Floquet periodic boundary conditions on the unit cell and sweep the k-space under COMSOL eigenfrequency sector. The Berry curvature is given by $\Omega_{\nu} = \nabla_k \times A_{\nu}$, where $A_{\nu} = -i\langle \mathbf{E}_{\nu}(k)|\nabla_k\mathbf{E}_{\nu}(k)\rangle$, with subscript $\nu$ representing either valleys K or K′, and $\mathbf{E}_{\nu}(k)$ represents the $\nu$-th eigenstate of the TE/TM band.

To obtain the edge bands between QSH and QVH domains, we consider a $40 \times 1$ supercell with Floquet periodic boundary, whose upper half is a rod and whose lower half is a pair of triangular prisms. $K_x$ is scanned from $-\pi/a$ to $\pi/a$ with 121 steps. Twenty eigenfrequency points around 21 GHz are calculated. Frequency points corresponding to edge states at the upper/lower edge of the supercell are removed.

The numeric transmission profiles in Figs. 3, 4 are obtained using the COMSOL frequency sector at 21 GHz. The boundary conditions are a perfectly matched layer as in Fig. 3a–c and a second-order scattering boundary in Fig. 4b, d.

**Measurements of edge modes and dwell time.** Spectra of the field transmission coefficient are taken using an Agilent PNA-X Network Analyzer. The source dipole is connected to the vector network analyzer via a power amplifier. Two sets of holes are drilled through the copper plates. One set with diameter of 1.1 mm, is for the insertion of an antenna, the other, with diameter of 3.3 mm, is for fixing the position of the rods and triangular prisms. The triangular prisms can be rotated and are fixed by screws. Both source and detector antennas are inserted into the holes in the copper plate to a depth of approximately 8 mm. Since the wave intensity oscillates along the interface, measurements of intensity are averaged over 20 points in a small region near the source and the output using 5 successive points on 4 parallel lines near the boundary. We apply a moving average to the transmission spectra to smooth out fluctuations. Each frequency point is replaced by the average value of the 15 nearest data points corresponding to a frequency interval of 3.75 MHz.

The phase of the electric field is calibrated by subtracting the measured phase from the phase measured with the source and detector antennas in close proximity in air. The spectral derivative of the phase, $\frac{d\varphi}{d\omega}$, is obtained from the average value of $\frac{\Delta\varphi}{\Delta\omega}$ over the bandgap (20.3–21.3 GHz), with $\Delta\omega = 7.5$ MHz.

**Data availability**. The authors declare that all data that support the findings of this study are available from Yuhao Kang at ykang1@gradcenter.cuny.edu upon reasonable request.

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

## Acknowledgements

The work of Y.K., X.C., and A.Z.G. is supported by the National Science Foundation (NSF/DMR/-BSF: 1609218). X.N. and A.B.K. are supported by the National Science Foundation (grants CMMI-1537294 and EFRI-1641069).

## Author contributions

Y.K. and X.C. performed the experiments, Y.K. analyzed the data, Y.K. and X.N. carried out the numerical simulations, Y.K., A.Z.G., and A.B.K. wrote the manuscript, A.B.K.

conceived the project, A.Z.G. contributed to the experimental design and direction of the project.

## Additional information

**Competing interests:** The authors declare no competing interests.

