## [Peer Review File · Nature Communications]

Reviewers' comments:

Reviewer #1 (Remarks to the Author):

In this manuscript, the authors present a study of the edge states created at the interface between a topological insulator and a valley photonic crystal. Independently, if either of these two composite systems were placed adjacent to a topologically trivial photonic crystal, some form of protected edge states would develop. However, by placing these two different classes of topological materials next to one another, additional control can be obtained over the edge states, and it is this additional control which is experimentally observed in this manuscript. I find this extra degree of freedom which is introduced by creating a hybrid of these two different types of crystals interesting. However, the structure studied in this manuscript was previously suggested by these same authors in one of their recent prior works, Ref. 33 in this manuscript, see Fig. 2b therein. As such, although I think that this manuscript is worthy of publication somewhere, I'm not convinced it meets all of the criteria for publication in nature communications, as this project strikes me as an interesting, but incremental development.

A few other quick comments:

In the discussion of the valley-dependent splitter, it would be useful to provide a quick discussion of the result of inputs into ports 2 and 3.

In Figure 3, which of these data are the result of simulation vs experiment? The caption is unclear.

Reviewer #2 (Remarks to the Author):

In this manuscript the authors report on the first experimental demonstration of a photonic system displaying 'hybrid' topological edge states that combine the valley and pseudo-spin degrees of freedom. They do that by creating an interface between a cm-scale valley photonic crystal (based on a triangular lattice of properly designed copper triangular prisms) and a topological photonic crystal (formed by a lattice of copper rods with concentric collars). By optimizing the geometrical parameters of the system at each side of the interface, they manage to overlap the bandgaps of the rod-collar-lattice and the triangular lattice. That in turn, enables the existence of spin-valley polarized edge states at the interface (i.e., edge states that combine both valley- and pseudospin DOFs). This entanglement of two degrees of freedom allows the authors to demonstrate a Y-type beam splitter operating as a valley filter. In addition, the authors present a study of the transport properties characterizing the studied edge states in presence of disorder. They also support their experimental results with full-wave numerical simulations.

The manuscript is well written and organized. The reported results are sound and convincing. However, I have an overall main caveat on this work. In my view, the main impact of the reported results is their eventual/future application to the optical regime, where, as the authors themselves nicely explained in the introduction of this manuscript, the concepts regarding valley DOF have their true impact beyond their fundamental interest (mainly because the concepts linked to topological designs are too involved to be fabricated at the nanoscale). I really do not see how one could 'scale down' the main ideas of this work to the optical regime. In other words, the advances demonstrated in this work seem to be crucially dependent on the particular realization reported by the authors. This realization, in turn, seems very specific of cm-scale systems. (Again, I am not questioning the fundamental value of the reported results, but their possible impact beyond that fundamental interest).

A careful and detailed response/discussion of the above point is needed. I would really appreciate if the authors do not reply with generic answers regarding this point, but rather with specific

directions or implementations that could make their findings applicable in the optical regime. (A small additional comment: the Methods section should be extended significantly, including the detailed specifics of how experimental measurements were made as well as the details of the simulations).

Reviewer #3 (Remarks to the Author):

In this work Kang et al. study pseudospin-valley locking along the interface between a valley photonic crystal and a pseudospin Hall photonic crystal. The authors use symmetry breaking approach to induce respectively gaps at the valley Dirac cones. They then move through the standard characterization of chiral edge modes at the interface between the materials, namely, by showing transport through the bulk and boundary, as well as, by studying the detrimental effect of noise. Last, the construction of a Y-junction between three metamaterials, two opposite valley crystals and a single spin Hall, allows for the generation of a valley-dependent beam splitter.

The field of topological photonics has gained much attention in recent years. Following various initial implementation of chiral edge propagation, the field is searching for new directions and applications. In this context, the current work follows well established paths of characterizing chiral edge modes, but at the same time offers a new touch to this endeavor in the form of a technological application. I would therefore recommend publication in Nat. Comm. once the following points were addressed:

1. The authors heavily rely on previous works to explain their employed techniques. It would be useful to keep this work self-contained and include some of the arguments and models that lead to the existence of valley Hall in the triangular crystals and spin Hall in the rod structure.
2. An additional example of missing information involves Fig. 3d, where for a wide readership, it would not be clear what role is played by the collars and their contact to the plate.
3. Considering that the main result of this work involves the description of what kind of edges would exist at the interface between a valley Hall and a spin Hall systems, there is insufficient detail as to why a hybridized single spin-valley locked chiral channel is the correct bulk-boundary correspondence in such a device.
4. The authors might want to refer to a recent review on topological photonics: arXiv:1802.04173 for introducing the field.

Response to Reviewers

Reviewer #1 (Remarks to the Author):

Reviewer #1: General remarks

In this manuscript, the authors present a study of the edge states created at the interface between a topological insulator and a valley photonic crystal. Independently, if either of these two composite systems were placed adjacent to a topologically trivial photonic crystal, some form of protected edge states would develop. However, by placing these two different classes of topological materials next to one another, additional control can be obtained over the edge states, and it is this additional control which is experimentally observed in this manuscript. I find this extra degree of freedom which is introduced by creating a hybrid of these two different types of crystals interesting. However, the structure studied in this manuscript was previously suggested by these same authors in one of their recent prior works, Ref. 33 in this manuscript, see Fig. 2b therein. As such, although I think that this manuscript is worthy of publication somewhere, I'm not convinced it meets all of the criteria for publication in nature communications, as this project strikes me as an interesting, but incremental development.

Authors' response to General remarks

We would like to thank Reviewer #1 for thoroughly reading our manuscript. Indeed, the novelty of the work stems from the combination of two distinct topological phases and the additional control of edge states in such configuration. Reviewer #1 underlines these novel aspects and advantages of the proposed scheme for controlling topological edge states. Nonetheless, Reviewer #1, suggests that the work is incremental since the structure studied was suggested by us in our prior work (Ref. 33 in the original manuscript and 34 in the present manuscript.) We would like to emphasize that our previous work had focused exclusively on the photonic quantum spin-Hall (QSH) phase. The control of the valley degree of freedom was not discussed. In the previous work, we studied edge states supported by the domain wall between two QSHE lattices, in which the valley degree of freedom could not be utilized. For this reason, we needed to assemble a new structure, only half of which is similar to one in Ref. 34; the second half represents a completely new structure, the spin-degenerate valley-Hall (VH) insulator. We would therefore like to emphasize that the structure we study is new and consists of a topological waveguide between QSH and VH crystals. The edge states are different from those studied in Ref. 34; in addition to exhibiting spin-polarization they are also valley-polarized. Moreover, as mentioned by the other reviewers, the present manuscript demonstrates for the first time a spin-valley locked filter and thereby brings the field of topological photonics and the whole concept of synthetic degrees of freedom one step closer to practical applications.

In view of the comments of Reviewer #1, we have revised the text in order to lay out the differences between the present work and Ref. 34. We hope that Reviewer #1 will appreciate this revision and find the paper suitable for publication in Nature Communications.

Reviewer #1: Specific remarks

A few other quick comments:

Remark 1

In the discussion of the valley-dependent splitter, it would be useful to provide a quick discussion of the result of inputs into ports 2 and 3.

Authors' response

We thank Reviewer #1 for this valuable suggestion. The corresponding discussion was added to the revised manuscript. We agree that having a more detailed discussion of the response when other ports are used clarifies the distinctive physics of valley-splitting.

Remark 2

In Figure 3, which of these data are the result of simulation vs experiment? The caption is unclear.

Authors' response

The caption of Figure 3 has been revised to clarify this point.

Reviewer #2 (Remarks to the Author):**Reviewer #2: General remarks**

In this manuscript, the authors report on the first experimental demonstration of a photonic system displaying 'hybrid' topological edge states that combine the valley and pseudo-spin degrees of freedom. They do that by creating an interface between a cm-scale valley photonic crystal (based on a triangular lattice of properly designed copper triangular prisms) and a topological photonic crystal (formed by a lattice of copper rods with concentric collars). By optimizing the geometrical parameters of the system at each side of the interface, they manage to overlap the bandgaps of the rod-collar-lattice and the triangular lattice. That in turn, enables the existence of spin-valley polarized edge states at the interface (i.e., edge states that combine both valley- and pseudospin DOFs). This entanglement of two degrees of freedom allows the authors to demonstrate a Y-type beam splitter operating as a valley filter. In addition, the authors present a study of the transport properties characterizing the studied edge states in presence of disorder. They also support their experimental results with full-wave numerical simulations.

The manuscript is well written and organized. The reported results are sound and convincing. However, I have an overall main caveat on this work. In my view, the main impact of the reported results is their eventual/future application to the optical regime, where, as the authors themselves nicely explained in the introduction of this manuscript, the concepts regarding valley DOF have their true impact beyond their fundamental interest (mainly because the concepts linked to topological designs are too involved to be fabricated at the nanoscale). I really do not

see how one could ‘scale down’ the main ideas of this work to the optical regime. In other words, the advances demonstrated in this work seem to be crucially dependent on the particular realization reported by the authors. This realization, in turn, seems very specific of cm-scale systems. (Again, I am not questioning the fundamental value of the reported results, but their possible impact beyond that fundamental interest).

A careful and detailed response/discussion of the above point is needed. I would really appreciate if the authors do not reply with generic answers regarding this point, but rather with specific directions or implementations that could make their findings applicable in the optical regime.

Authors’ response to General remarks

We are thankful to Reviewer #2 for the positive evaluation of our work. We agree with the reviewer that the combination and entanglement of different DOF is of great fundamental interest and may change the way we control electromagnetic radiation and light. Indeed, bringing these ideas into the optical domain can be extremely beneficial for telecommunications and optical signal processing. This would entail a new design that avoids using metals since Ohmic losses in the optical domain would be unacceptable. Recently, we proposed an all-dielectric designs for QSHE systems [Nature Photonics volume 11, pages 130–136 (2017)] and we are currently working on their experimental implementation at microwave frequencies [arXiv:1705.07841]. However, all these designs are based on dielectrics with dielectric properties ($\epsilon \sim 40$) that are not feasible in the optical domain.

Nonetheless, following the request of Reviewer #2, we have been able to design a QSHE system which is based on germanium ($\epsilon = 16$) whose fabrication is feasible with the use of conventional nanofabrication techniques. The structure that we obtained represents a triangular array of germanium nano-disks with dimensions tuned such that the array supports two doubly degenerate Dirac cones stemming from electric and magnetic dipolar modes at both the K and K’ points of the Brillouin zone. The band structure plotted in Supplementary Figure 1 and reproduced below exhibits two pairs of Dirac bands whose degeneracy effectively emulates the spin degree of freedom. In order to introduce coupling between electric and magnetic modes, the magneto-electric coupling which gives the bianisotropic response, we reduce the out-of-plane inversion symmetry by introducing a circular notch on one of the flat faces of the cylinders. This gives rise to a complete photonic band gap, as shown in the figure below. The resultant top and bottom bands do not cross anywhere in the Brillouin zone. This implies that the topological properties are completely defined by the hybridization of the magnetic and electric Dirac bands near the K and K’ points. Our preliminary calculations for this optical design neglect the effect of the substrate, since the structure is fabricated on top of a 20-nm-thick SiN membrane. Further advances towards a more easily fabricated structure can be achieved through the use of layers of materials with higher permittivity in the visible frequency range, such as silicon and germanium.

Supplementary Figure 1| Band structure of all-dielectric optical metamaterials. (left panel) Numerically calculated photonic band structure of spin-valley degenerate germanium-based all-dielectric optical metamaterial. (right panel) Photonic band structure of optical metamaterial design with magneto-electric coupling. Insets in panels show the corresponding dielectric nano-disks of the two metasurfaces. The light-green shaded area illustrates the spectral bandwidth of the topological band gap. The position of the radiative continuum (above the light cone) is marked by the blue shaded areas. The geometrical parameters are as follows: the period is 700 nm, the large disk radius and height are 263 nm and 168 nm, the small disk radius and height are 188 nm and 68 nm, respectively.

We are pursuing further efforts to go beyond the QSHE to implement entangled valley-spin states similar to those reported in the present manuscript. However, for the present, we hope that the preliminary results presented in the new Supplementary Information section will convince Reviewer #2 that it is reasonable that the ideas proposed here in the context of microwave metamaterials can be implemented in the optical domain. This will undoubtedly require much more challenging fabrication methods such as electron beam lithography and etching performed on the nano-scale.

We agree that the question of the feasibility of the proposed effects in the optical domain is of great importance. We have therefore added a section to the Supplementary Information of the revised manuscript describing our progress thus far.

Reviewer #2: Additional comments

A small additional comment: the Methods section should be extended significantly, including the detailed specifics of how experimental measurements were made as well as the details of the simulations.

Authors' response to Additional comments

Following the recommendation of Reviewer #2, the Methods section has been significantly expanded to include detailed information regarding the experimental measurements.

In conclusion, we would like to thank Reviewer #2 again for the supportive comments and valuable suggestions, which, we believe, have allowed us to improve the quality and relevance of the work.

Reviewer #3 (Remarks to the Author):

Reviewer #3: General remarks

In this work Kang et al. study pseudospin-valley locking along the interface between a valley photonic crystal and a pseudospin Hall photonic crystal. The authors use symmetry breaking approach to induce respectively gaps at the valley Dirac cones. They then move through the standard characterization of chiral edge modes at the interface between the materials, namely, by showing transport through the bulk and boundary, as well as, by studying the detrimental effect of noise. Last, the construction of a Y-junction between three metamaterials, two opposite valley crystals and a single spin Hall, allows for the generation of a valley-dependent beam splitter.

The field of topological photonics has gained much attention in recent years. Following various initial implementation of chiral edge propagation, the field is searching for new directions and applications. In this context, the current work follows well established paths of characterizing chiral edge modes, but at the same time offers a new touch to this endeavor in the form of a technological application. I would therefore recommend publication in Nat. Comm. once the following points were addressed:

Authors' response to general remarks

We thank Reviewer #3 for their very encouraging remarks and positive evaluation of our work, and for the recommendation that the manuscript be published. We very much appreciate Reviewer #3 comment on technological applicability of the proposed valley-spin entanglement. In the revised manuscript, we have implemented all the changes requested by the reviewers. We believe this has improved the clarity of the manuscript and made it more self-contained.

Reviewer #3: Additional remarks

Remark 1

1. The authors heavily rely on previous works to explain their employed techniques. It would be useful to keep this work self-contained and include some of the arguments and models that lead to the existence of valley Hall in the triangular crystals and spin Hall in the rod structure.

Authors' response

Following Reviewer #3 suggestion, we have provided additional details on the physics of pseudo-spin and valley DOF in our system in the Supplementary Information. In addition, we now provide a description of the effective Hamiltonian of the system, including pseudo-spin and valley potentials generated in the crystal by the reduction of symmetries. We believe that the revised manuscript together with the Supplementary Information is now self-contained.

Remark 2

2. An additional example of missing information involves Fig. 3d, where for a wide readership, it would not be clear what role is played by the collars and their contact to the plate.

Authors' response

We would like to thank Reviewer #3 for this suggestion. Indeed, it is sometimes hard for the authors to see which information should be included in the text. We have now expanded the description of the structure and the role of collars in the symmetry breaking which are required to generate the pseudo spin-orbital potential. This should facilitate the understanding of electromagnetic states supported by the chosen geometry. This will also help readers understand the difference between cases of weak and strong disorder, when collars are placed between the copper plates and when the collars make contact with the opposite plate, respectively .

Remark 3

3. Considering that the main result of this work involves the description of what kind of edges would exist at the interface between a valley Hall and a spin Hall systems, there is insufficient detail as to why a hybridized single spin-valley locked chiral channel is the correct bulk-boundary correspondence in such a device.

Authors' response

This is an excellent question which merits further explanation. The answer is now given in the revised manuscript and is mathematically elaborated in a new section of the Supplementary Information. There we explain that the structure of the photonic modes ensures that both valley and spin potentials result in non-vanishing Berry curvature at each valley for each spin. Thus, for the valley crystal, the Chern number is valley specific with a magnitude of $1/2$, but with opposite signs at the two opposite valleys for each of the spins. For the QSH crystal, on the other hand, each of the valley contributions has the same sign ($1/2$ for spin-up and $-1/2$ for spin down at both valleys). The bulk-interface correspondence should then be applied independently at each valley. The number of modes equals the difference in the spin and valley Chern numbers at a particular valley, and the sign agrees with the sign of the difference. Using this principle, one can immediately predict both the presence and direction of propagation of the edge state at a

particular valley. This argument is further supported by the effective Hamiltonian description in the Supplementary Information.

Remark 4

4. The authors might want to refer to a recent review on topological photonics: arXiv:1802.04173 for introducing the field.

Authors' response We were happy to cite this article it in our revised manuscript.

We thank the reviewer for pointing to this new comprehensive review on topological photonics.

We thank all the reviewers for their important comments which have improved the paper. Changes made in the main text are highlighted in yellow and a new Supplementary Information section has been added.

REVIEWERS' COMMENTS:

Reviewer #1 (Remarks to the Author):

The authors have addressed my concerns, and upon rereading and spending some time considering the manuscript, I recommend its publication in its present form.

Reviewer #2 (Remarks to the Author):

The authors have successfully addressed my main caveat regarding the generalisation to the optical regime of the approach introduced in their work.

Reviewer #3 (Remarks to the Author):

The authors have answered my critique in full and I can now support publication of the manuscript in Nature Communication.